# Physical Healthcare for People with a Severe Mental Illness in Belgium by Long-Term Community Mental Health Outreach Teams: A Qualitative Descriptive Study on Physicians’, Community Mental Health Workers’ and Patients’ Perspectives

**DOI:** 10.3390/ijerph20010811

**Published:** 2023-01-01

**Authors:** Nicolaas Martens, Eline De Haeck, Evelyn Van De Vondel, Marianne Destoop, Kirsten Catthoor, Geert Dom, Kris Van Den Broeck

**Affiliations:** 1Collaborative Antwerp Psychiatric Research Institute (CAPRI), University of Antwerp, B-2610 Antwerp, Belgium; 2Multiversum Psychiatric Hospital, B-2530 Boechout, Belgium; 3Department of Nursing, Karel de Grote University of Applied Sciences, B-2018 Antwerp, Belgium; 4Family and Population Health (FAMPOP), University of Antwerp, B-2610 Antwerp, Belgium; 5ZNA Stuivenberg Psychiatric Hospital, B-2060 Antwerp, Belgium

**Keywords:** community mental healthcare, mental health nursing, integrated care, organization of healthcare, physical healthcare, severe mental illness

## Abstract

Background: People with a severe mental illness (pSMI) often have comorbid physical health problems, resulting in a lower life expectancy compared to the global population. In Belgium, it remains unclear how to approach health disparities in pSMI in a community setting. This study explores the perspectives of both care professionals and patients on physical healthcare in Belgian community mental services, aiming to identify good practices, barriers and points of improvement. Methods: An exploratory qualitative design that used a semi-structured focus group interview with physicians combined with individual face-to-face interviews with physicians, mental health professionals and patients. Results: We identified care professional-, patient-related and organizational factors, as well as points of improvement. The identified themes linked to care professionals were communication, task distribution, knowledge, time and stigmatization. The co-location of services was the main theme on an organizational level. Conclusions: As community-based mental health services in Belgium emerged in the past decade, addressing physical health in pSMI is still challenging. Our findings suggest that there is a need for improvement in the current healthcare provision. Multidisciplinary guidelines, shared patient records, enlarging nurses’ tasks, providing financial incentives and a structural integration of primary and psychiatric care were perceived as major points of improvement to the current Belgian healthcare organization.

## 1. Introduction

Severe mental illnesses (SMI) include severe and persisting psychiatric disorders that impede global functioning and require treatment by mental health services for two years or more. SMI mostly encompass psychotic or bipolar disorders but, if the criteria regarding the duration of treatment and global functioning are met, other conditions could also be perceived as an SMI [1]. The deficient functioning of persons with an SMI (pSMI) could (re)induce psychiatric symptoms, resulting in a circular, symptom-enhancing process [2].

With regard to physical health, when compared to the general population, the life expectancy of pSMI is 13–30 years lower [3,4,5]. Cardiovascular problems underlie most of the premature deaths in pSMI due to multiple known risk factors such as higher tobacco use and higher prevalence of diabetes, dyslipidemia, hypertension, and obesity. These risk factors can be negatively influenced by the use of psychotropic medication. In addition, pathogenic mechanisms such as inflammation might be common factors underlying both mental and physical disorders [6,7,8,9].

In general, intensive care coordination or network-based community care for pSMI is recommended.

As in many countries, an increasing number of pSMI in Belgium are receiving mental health treatment through community mental health teams for long-term care (CMHTltcs) in their personal home settings. As the mental health reform had a late onset compared to other countries, up until now community mental health services have been in development [2]. With regard to physical health for pSMI, previous research showed that CMHT-ltcs currently provide psychosocial services, with little attention to somatic health disparities [10,11].

A recent Belgian study, examining the perspectives of both patients and care professionals regarding physical healthcare, revealed that inpatient infrastructure and staff were insufficient, thereby adding to the gap between the increased risk of cardiovascular problems in pSMI and access to appropriate care [12]. In primary care, barriers contributing to inappropriate risk increase and treatment have also been identified, such as the fact that pSMI tend to postpone primary care visits, limited interprofessional communication, policy-related financial barriers and stigma among professionals [13,14,15]. In addition, both stigmatization and self-stigmatization increase the barriers to proper physical healthcare [13,16,17].

As it remains unclear how to tackle the forementioned health disparities in pSMI treated by CMHTltcs, this study aimed to explore the perspectives of both care professionals and patients on physical healthcare in Flemish community mental services. The goal was to identify good practices, barriers and implementable points of improvement in addressing physical health.

## 2. Materials and Methods

### 2.1. Design

An exploratory qualitative design was used to examine the research questions. We conducted a semi-structured mixed focus group interview with both general practitioners (GPs) and psychiatrists active in community practices and who had regular contacts with pSMI. Individual face-to-face interviews were also organized to explore the perspectives of physicians who could not attend the focus group due to the COVID-19 measures at that time. Patients and care professionals (other than physicians) working in a CMHTltc were interviewed individually. Individual interviews used a predetermined semi-structured topic list.

### 2.2. Participants

#### Inclusion Criteria

-Physicians: both psychiatrist and GPs were included if they had pSMI in their caseload as well as an ambulatory, outpatient practice.-Patients: patients were eligible for inclusion in this study if they had an SMI and were in treatment in an CMHTltc. Patients had to be 18 years or older, without an intellectual disability and speak the Dutch language fluently.-Care professionals of a CMHTltc: care professionals were included if they worked in a CMHTltc.

Some psychiatrists were linked to both a community practice and an outreach team and were assigned to a double background, if the inclusion criteria of both subgroups were met.

### 2.3. Recruitment Strategy

To recruit the physicians (i.e., psychiatrists and GPs), a convenience sample was invited by e-mail. We contacted (professionals working at) psychiatric and primary care services and asked those who reacted whether they knew additional potential candidates (snowballing). The care professionals working in a CMHTltc were recruited using a convenience sample by sending an invitation the CMHTltcs. The patients were recruited by sending an invitation letter to the patients within the caseload of two CMHTltcs.

### 2.4. Ethical Approval

Ethical approval was given by the Ethical Committee of Antwerp University Hospital, with study numbers 19.50.616 an B-300.2021.000045. All participants, both patients and care professionals, provided a written informed consent prior to enrollment in the study.

### 2.5. Data Collection and Analysis

Based on the literature, three separate interview scripts were developed for physicians, care professionals in a CMHTltc and patients of a CMHTltc, respectively. Overall, the interviews were focused on identifying barriers, facilitators, personal beliefs and points of improvement, based on the following questions:-How do participants experience the quality of physical healthcare for pSMI?-What are facilitators and barriers in the collaboration between different caregivers and the patients regarding physical healthcare in a CMHTltc?-Are there good practices and suggestions concerning physical healthcare for pSMI in a CMHTltc?

These questions were used throughout all the interviews that were conducted. As semi-structured interviews were conducted, the content of previous interviews could influence the questioning in the subsequent interviews, without ignoring these three main questions. This iterative approach aimed to attain more in-depth information from the participants. A focus group with psychiatrists and GPs was, due to local COVID-19 measures, organized online using Cisco Webex and was guided by a moderator (the author EDH). Video recordings of the focus group were viewed afterwards by the observer (the author NM). For the physicians, further data collection was organized using individual interviews, as some of them were not able to attend the focus group. Following transcription, inductive coding and thematic analysis of the interviews were executed independently by authors EDH and NM. In a second step, the codes were compared and clustered together, resulting in a hierarchical code tree. The findings were described based on these clusters of codes.

The care professionals of the CMHTltcs were interviewed by author NM, who also performed inductive coding and further thematic analysis after transcription. The interviews of the patients treated by a CMHTltc were conducted by author EVDV. Following transcription, these interviews were inductively coded and independently thematically analyzed by authors EVDV and NM. All transcripts were coded using Nvivo 12.

As several professional disciplines were questioned, to avoid interpretation bias, the background of the researchers was diverse (nurse, psychologist, psychiatrist, GP in training). The emerging themes were listed, and the results were reviewed by all authors. All original recordings were deleted after the analysis.

Next, the transcripts were coded by two KCE researchers (LK and WC) with NVIVO software, version 1.5.1. The data were analyzed by thematic analysis.

## 3. Results

Twelve physicians participated in the study, of which eight were in the focus group, and six were interviewed individually. Two physicians were interviewed individually after they participated in the focus group. Fourteen patients from two CMHTltcs and four care professionals of a CMHTltc were interviewed individually (for an overview, see Table 1).

### 3.1. Care Professional-Related Factors

#### 3.1.1. Communication

Communication was perceived as a main need of improvement by all participants. Among the GPs and psychiatrists, communication mostly occurred using low-quality letters of discharge or referral, often hindering a timely report on the patients’ status. For outpatient consultations, the psychiatrists resulted not to share information with the GPs, causing the psychiatric reports not to be consultable on electronic health platforms by the GP. Although lab results, medication lists and the GP’s summary of each patient’s general health status were available, some psychiatrists never consulted them. In the CMHTltcs, electronic medical records were sometimes unavailable; therefore, communication often relied on time-consuming telephonic contacts.

Although opinions differed, most patients indicated that there was little to no communication between the care providers involved. Some patients indicated that this was tiring, because it required a lot of self-management, and they sometimes lacked the necessary support in this. Other patients found it very important to maintain their self-management.

*“The outreach team and the GP have no contact with each other either. I’m very sorry about that! With my pneumonia too, well…. Actually a multidisciplinary collaborative care team with a psychiatrist, outreach team, GP…that’s a must…that’s normal, isn’t it? At the moment, these are actually all separate parts.”* (P7)

Among the physicians, knowing one another personally was found to improve their professional collaboration. Especially the GPs found it helpful to personally know a psychiatrist because it would facilitate their communication. A few psychiatrists stated to find it helpful to know the GP as regards somatic monitoring and follow-up. The physicians were frustrated concerning their mutual accessibility due to differences in working hours, unclarity of availabilities and not answering telephone calls during consultations or therapy sessions.

*“…in case of admission, we receive a letter or after ending an ambulatory trajectory…but for those people that have a chronic mental health condition, we often receive no letters”* (GP 7)

GPs reported to expect access to a correct medication list and clear communication about somatic problems, such as abnormal blood results, with an active referral of the patient to the GP. There was agreement between the physicians that major changes, problems or medication adjustments should always be communicated. The psychiatrists expect notifications from the GP in case of new important (somatic) problems. In general, cooperation and communication between GPs and psychiatrists was perceived as being variable and dependent on the care professional or instances involved.

#### 3.1.2. Task Distribution

All participants agreed that for CMHTLtcs, the task of physical follow-up should mainly be reserved to GPs, because of their holistic approach and ability to offer more continuity in care. The GPs, however, felt that the psychiatrists should provide more information and instructions. The psychiatrists agreed to this opinion but would expect more initiative from the GPs. A need for more specific agreements emerged.

*“Yes, I assume that the part of physical care is mainly our task anyway. I’m assuming that’s not really for the psychiatrist. It is easier to assume that it is for the psychiatrist to keep the psychiatric medication good, (…) I think that is exactly okay because that is something we can do well, which we are used to do.”* (GP 7)

Most of the patients in the CMHTltcs indicated that they experienced both psychological and physical complaints. In general, the patients indicated that the CMHTltc supported them in psychosocial matters and did not consider the outreach team as a point of contact regarding physical complaints. The patients appeared to consider the GP as the central contact for physical complaints, when present (there was no regular GP for two respondents). The psychiatrist was perceived as a care provider regarding psychological complaints.

#### 3.1.3. Knowledge

The GPs reported to usually screen pSMI according to the “standard screening schedule” as provided by the national guidelines in primary care. If any visible risk factors are observed, screening tests are performed more quickly. Not every GP was aware of the existing physical health guidelines for pSMI and used those for the general population.

The psychiatrists emphasized that the psychosocial approach was limited, and the knowledge of physical health referral points (e.g., dietician, exercise coaching) was rather scarce. One psychiatrist even indicated to feel a certain resistance to perform “medical” acts, such as blood pressure or abdominal circumference measurements. The care professionals in a CMHTltc also indicated having little knowledge of physical health and limiting their services to psychosocial support.

*“(...) if I see an elevated cholesterol level, I first approach the patient about dietary measures, (…), I automatically refer to the GP. I’m never going to prescribe a statin myself, I don’t even think about it.”* (Psy 1)

#### 3.1.4. Lack of Time

When examining the possible barriers that would impede interprofessional collaboration, most GPs mentioned lack of time as a major barrier. In addition, the GPs experienced a rather limited return on their investment of time, evoking feelings of despondency. The GPs often felt that they had to set priorities and that physical health was not addressed in patients with major or acute psychosocial problems.

*“(…) I also recognize the time pressure under which we work and that you really have to see which complaints needs to be handled first and ‘what should I do to get the greatest return’.”* (GP 2)

#### 3.1.5. Stigmatization and Diagnostic Overshadowing

The patients reported having to wait a long time before receiving physical healthcare or not being taken seriously by the health care providers. The care professionals tended to rule out possible psychological explanations first. Some patients indicated that they were sufficiently assertive to sort this out themselves, but for most clients this required a lot of effort.

*“No…I once had an experience with the outreach team…When I was with my girlfriend, I hurt myself playing sports or something…and the outreach team said: ‘have a seat, and rest’…That’s all they said! They didn’t say: ‘wouldn’t you go to a physician, or wouldn’t you take an X-ray?’. I had to do it all by myself.”* (P3)

Overall, the patients appeared to understand the need for their psychotropic medication, but when reporting physical complaints to the psychiatrist related to medication use, some patients experienced that their complaints were minimized and ignored. This could place patients in a difficult impasse. The blood levels were not always strictly checked by the psychiatrist, some clients mentioned they had to remind the psychiatrist themselves whenever it was necessary. Several patients indicated that psychological help was accessible, though difficult to acquire in case of physical complaints, evoking frustrations, as appropriate help was only obtained with much effort from the patient.

*“The psychiatrist was already aware, the GP was already aware. And he always gave me medication, like: ‘we’re going to try that once, we’re going to try that once’. And that didn’t help, but in the long run…that was impossible! I was limping. I couldn’t walk normally anymore. And it hurt…Yes, it took a month before I was diagnosed with thrombosis.”* (P10)

### 3.2. Patient-Related Factors

#### 3.2.1. Patient Characteristics

According to the physicians, an adequate follow-up mostly depended on patient-related factors and feasibility. A strict implementation of the guidelines was not always successful due to resistance or lack of motivation of the patient, implying a tailored approach on the level of the patient. In addition, the patients sometimes did not have a GP or did not attend the consultations. The professionals in a CMHTltc also underlined the complexity of the cases in which they were in involved, where priorities were mostly set on social services, such as housing and financial problems. The patients of the CMHTltc experienced more difficulties to initiate treatment with regard to physical health problems and remained connected with the professionals involved.

*“I realize that, when it concerns my own patients, I don’t give any attention to physical health. My patients often are hard to reach, on the edge of admission…so physical health is not a priority…Also, I often don’t know which medication my patients are taking.”* (CP 1)

*“For me, psychological help is easier to get, yes it is. If I’m panicking for some reason…I can call right away. They pick up immediately and they calm me down, and say: ‘should I come by soon?’ or ‘I will call back later’”* (P14)

Limited insight in their illnesses was also considered a barrier on a patient level, and the patients sometimes preferred that information about their psychiatric condition was not shared with primary care services or did not find this important, which could impede a good follow-up.

*“Yes, a GP is mainly for the physical…and he explained everything in detail, a little too well. And at a certain moment he wanted to adjust my medication, especially my sleeping medication…‘we will have to take a closer look at that’…then I ran away and slammed the door”* (P10)

#### 3.2.2. Care Organization

Two clients received care from a multidisciplinary community primary-care health center. These are practices reimbursed by capitation and involve a multidisciplinary team; the patients found this very positive. Especially, the unambiguous approach and accessibility were mentioned, which gave the patients a feeling of receiving timely and adequate help. When the GPs were working in solo practices, referrals towards other services were perceived as a barrier for the patients, potentially resulting in additional care-related costs.

*“I have experience with outpatients at some GP practices where a nurse is working…that actually works very well. He then often administers medication, provides follow-up of parameters and lab results. Of course, I don’t know to what extent the GP should play a role in that, I don’t have a good idea how that works in practice, because at the moment the nurses don’t have that much mandate either to function independently.”* (Psy 4)

### 3.3. Points of Improvement

On a patient level, it was agreed that setting achievable, adjusted goals for each patient would improve the chances of success.

On a meso level, the physicians agreed that regional activities in small groups and focused on caregivers operating in the same region could contribute to a better collaboration, e.g., getting to know each other in a more informal manner could contribute to a better collaboration.

Another proposal was the involvement of nurses in screening and education as an added value, according to the participating GPs. A nurse could dedicate more time to the patients, and this could add to the quality of care. A counterargument stated that there was already a shortage of nurses in hospitals.

*“Because I think that the nurses could well take on that follow-up, if it is not too complex, as a kind of case manager (…) the nurse knows very clearly what needs to be followed up in terms of lab, parameters, weight, all those things and that he or she can follow, in collaboration with the GP”* (Psy 4)

In CMHTltcs, nurses are currently present as team members, with mental health care as their primary task. The care professionals in the CMHTltcs are rather hesitant in allowing these nurses to address physical health disparities in their caseload, and this should be initiated in primary care services. A nurse, working both in primary care and in a CMHTltc, however, could be of added value in facilitating collaboration, as primary care services are perceived as difficult to access by the CMHT.

*“I don’t feel a nurse of the CMHTLTC should be responsible for somatic screening and follow-up of the team’s caseload, as it could increase the fragmentation of our services instead of integrating them. (…) but the actual care should be provided in primary care. Also, primary care is difficult to access for us as a team…”* (CP3)

On a macro level, firstly, the need for a simple and efficient (digital medical record) system of communication was mentioned. In addition, a specific guideline should be available to guide a tailored physical care delivery for pSMI. One psychiatrist proposed to organize outpatient mental health services in a more integrated manner and provide co-located physical and mental health services.

*Aa consensus document in Flanders, in Dutch, calibrated to what is reimbursed to us by the government for screening, examinations and consorts, I think that is a gap (…). Almost nothing happens somatically in the outpatient department. What is an outpatient clinic in most hospitals?...There are a few psychiatrists and if all goes well a few psychologists who do conversations (…)…But I think the integration of a somatic discipline in it or the accessibility to one is currently missing and needed.”* (Psy 5)

The financing of somatic care for mental health patients in Belgian health settings was found to be inadequate. According to several psychiatrists, the paucity of professionals and materials addressing physical health care is the worst in mental health community or outreach teams. More structural and financial support to increase integration of services was asked for, mostly by the physicians. One respondent raised the idea of an SMI care trajectory, with the advantages of affordable, more structured care and more educational options.

*“Yes, within the outreaching crisis resolution team, but also within the long-term care teams, this is not a priority and there is also a lack of setting or material in people’s homes or for people on the street. And I notice that this is often not a priority for policy makers.”* (Psy 4)

The patients requested to be taken more seriously by the professionals when formulating symptoms and to look beyond the psychiatric illness. The centralization of various services would be an added value for approximately half of the patients that were interviewed. Multidisciplinary community primary-care group practices could be of added value in lowering barriers, but it was noted that there are too few of them. The fact that there is continuity of care in these specific types of primary care centers appeared reassuring for the patients.

*“…taking the patient’s question seriously. It’s not because we’re confused…I had a friend like that…He was in a lot of pain and needed surgery. But caregivers did not listen because they thought he was not psychologically well at the moment, …”* (P14)

In addition, a more structural collaboration could ensure that patients are viewed in a more ‘holistic’ way and that there is more clarity for everyone. The outreach team could possibly act as a point of contact in this regard.

*“The outreach team, I think they do a very good job. It’s all hard to manage, isn’t it? (…) I would use one contact person, your contact person. A kind of personal guide to whom you can just say: ‘I’ve got this and this and this on my mind, you just make sure it gets to the right person’”* (P2)

## 4. Discussion

We conducted a qualitative study among GPs, psychiatrists, patients and CMHTLTC team members in an urban region in Flanders, Belgium, to explore their perspectives on physical health care delivery for pSMI in community healthcare.

We found that physical healthcare delivery for pSMI is experienced as suboptimal, despite the availability of various guidelines in the international literature with regard to screening and monitoring [13,18,19,20,21,22,23]. Importantly, a clear description about multidisciplinary task distribution is often lacking in these studies. We conclude that the current practice is characterized by few agreements on the task division between caregivers, often using ‘ad hoc’ approaches. The interventions are not adapted to the SMI population, and the available (inter)national guidelines seem to be rarely or not applied in an outpatient setting.

Previous research already highlighted difficulties in collaborative approaches for persons with depression in a Belgian context. pSMI often have complex comorbid problems and are more difficult to reach, especially in a context of community healthcare. A multidisciplinary guideline tailored to this target group and to the context of Belgian healthcare can be valuable in supporting healthcare professionals. To assure an evidence base of the guideline, using the GRADE framework could increase implementation and use in clinical practice [3,24]. International consensus highlights the necessity of providing a clear framework when addressing physical health in pSMI, of which shared care planning is an essential building block [25,26,27].

We identified several complicating factors in the Flemish context: barriers in communication caused by patient-, physician- and CMHTltc-related factors; physician-related factors such as lack of time, stigmatization and hesitance of the psychiatrists towards physical healthcare; patient-related factors such as high no-show rates and low medication adherence.

All participants indicated that it would be ideal for the GP to take charge of physical healthcare in the outpatient setting, requiring clear communication and more concrete agreements. To improve collaboration and communication, the co-location of services or community liaison services should be recommended. Previous research demonstrated this could improve physical health care screening and delivery [28,29,30].

Essential physical health checks for pSMI and the timing of when they should be performed are well-described in existing research [31,32]. Although some national Belgian guidelines are already adapted to target groups such as pSMI, a recent Belgian study underlined the need for a national multidisciplinary guidance on this topic, assuring an evidence base for pSMI, such as initiating somatic screening on a younger age compared to the general population [13,33,34]. This could provide guidance and structure and, in the case of an SMI “care trajectory”, even define financial aspects, as more financial efforts are desired. The majority of the participants involved in our study showed an interest in clarifying cooperation agreements and developing regional guidelines, which could be an area of interest for further research concerning this matter.

Firth et al. (2019) also emphasized the importance of efficient interdisciplinary care pathways and referral options, from both primary and secondary care, and specialized services [26].

We suggest that the role of nurses could be enlarged and concretized in care plans or protocols. Several guidelines, for example, predominantly assign cardiovascular screening in pSMI to (mental health) nurses [22,35]. In practice, however, this is not done in an organized manner. In Belgium, recently, the nurses have been working more commonly in general practices, and research shows several positive effects of the implementation of primary care nurses in the management of patients with chronic conditions and collaboration in practice, after receiving sufficient training and education [36].

In this study, the patient factors were important for the successful application of physical healthcare. Firstly, overcoming stigmatization and diagnostic overshadowing, by means of education and sensibilization, could increase the accessibility to healthcare services for the patients [37]. In addition, previous research showed that the use of peer navigators, i.e., pSMI supporting peers in adequately consulting healthcare providers, could increase patients’ consultation rates and adherence [16,38,39,40,41].

Our findings are limited to a small region in Belgium; therefore, the results should be interpreted with caution in terms of applicability in a broader, national perspective. In addition, given the fact that representatives of several disciplines were interviewed, the sample size for each discipline group was rather small, especially for professionals working in CMHTs.

To our knowledge, this is the first study examining physical healthcare for pSMI in a Belgian, outpatient mental health community context. We consider a strength that we included the perspectives of both patients and care professionals and mapped out the difficulties and suggestions openly and constructively. As this study included participants of different backgrounds, it is difficult to state that data saturation was achieved. In addition, due to the diversity of the researchers (nurse, psychologist, psychiatrist, GP in training), we may have reduced the risk of interpretation bias.

## 5. Conclusions

Both community-based mental health services and primary care services in Belgium experience difficulties in addressing physical health in pSMI. The patients of CMHTs also encounter several barriers in finding appropriate care for physical health problems. Our findings suggest that there is a need for improvement in the current healthcare provision. The development of (a) multidisciplinary guideline(s) could promote better collaboration and task distribution, as well as shared patient records. In addition, enlarging nurses’ tasks could increase the quality of physical health screening and follow-up. Concerning the current Belgian organization of healthcare, financial incentives and a structural integration of primary and psychiatric care were perceived as major points of improvement. The development of guidelines adapted to the Belgian context could improve health care delivery.

## Figures and Tables

**Table 1 ijerph-20-00811-t001:** Overview of the participants’ characteristics. (X = present at the focus group and/or interview).

Respondents	Sex	Age	Professional Context (If Caregiver) or Psychiatric Diagnosis (If Patient)	Focus Group	Interview
Psychiatrists					
Psychiatrist 1 (Psy1)	M	61	Solo practice/Psychiatric hospital/Outreach team	X	
Psychiatrist 2 (Psy2)	F	63	Solo practice/Psychiatric hospital/Outreach team	X	X
Psychiatrist 3 (Psy3)	F	25	Psychiatric hospital Outpatient consultation (resident)	X	
Psychiatrist 4 (Psy4)	F	39	Psychiatric hospital/Outreach team		X
Psychiatrist 5 (Psy5)	M	62	Solo Practice/Psychiatric hospital		X
General practitioners					
GP 1 (GP1)	F	46	Shared practice, fee-for-service	X	
GP 2 (GP2)	M	47	Group practice, fee-for-service	X	
GP 3 (GP3)	F	63	Group practice, fee-for-service	X	X
GP 4 (GP4)	F	54	Group practice, payment by capitation	X	
GP 5 (GP5)	F	51	Group practice, fee-for-service	X	
GP 6 (GP6)	F	28	Group practice, fee-for-service		X
GP 7 (GP7)	F	44	Group practice, fee-for-service		X
Community mental health team—professionals					
CMHTltc Care professional 1 (CP1)	F	27	Psychological consultant		X
CMHTltc Care professional 2 (CP2)	F	49	Occupational worker		X
CMHTLTC Care professional 3 (CP3)	F	43	Psychologist		X
CMHTLTC Care professional 4 (CP4)	F	39	Nurse		X
Community mental health team—patients					
CMHTLTC Patient 1 (P1)	F	34	Borderline personality disorder		X
CMHTLTC Patient 2 (P2)	F	33	Bipolar disorder		X
CMHTLTC Patient 3 (P3)	M	47	Psychotic depression		X
CMHTLTC Patient 4 (P4)	M	29	Schizophrenia		X
CMHTLTC Patient 5 (P5)	M	36	Schizoaffective disorder		X
CMHTLTC Patient 6 (P6)	M	51	Schizoaffective disorder		X
CMHTLTC Patient 7 (P7)	M	64	Substance abuse (polytoxicomania)		X
CMHTLTC Patient 8 (P8)	M	56	Schizoaffective disorder		X
CMHTLTC Patient 9 (P9)	F	42	Borderline personality disorder		X
CMHTLTC Patient 10 (P10)	M	57	Dual diagnosis, schizophrenia/substance abuse		X
CMHTLTC Patient 11 (P11)	F	53	Schizophrenia		X
CMHTLTC Patient 12 (P12)	M	38	Schizoaffective disorder		X
CMHTLTC Patient 13 (P13)	F	48	Bipolar disorder		X
CMHTLTC Patient 14 (P14)	M	41	Schizoaffective disorder		X

## Data Availability

Data can be provided on request by the corresponding author.

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
