# Peer review of "Physical Healthcare for People with a Severe Mental Illness in Belgium by Long-Term Community Mental Health Outreach Teams: A Qualitative Descriptive Study on Physicians’, Community Mental Health Workers’ and Patients’ Perspectives"

_ijerph, 2023, doi:10.3390/ijerph20010811_

Round 1
Reviewer 1 Report
Overall, I think the article is interesting and worth publishing.
I do have a couple of issues.
In summary, I would say:
- Concept should be better defined
- Discussion should be improved
- Some writing errors
- Somewhat odd conclusion
- You are very vague in terms of what analysis you used
Introduction
I am somewhat confused by the definition of SMI: “SMI mostly encompasses persons with a psychotic or bipolar disorder, but depending on the impact of the disorder, other conditions could also be perceived as a SMI”. So… Anything can be an SMI. Post-traumatic stress disorder, depression, narcissistic personality disorder…
Does the “severe” refer to the conceptualization: e.g. PTSD is more severe than adjustment disorder, or to the experience of the disorder?
That insures that the rest of the introduction really should apply to all “severe” disorders, I would assume. Though reference 5 does only refer to the earlier mention psychotic disorders and bipolar disorders.
My point is: I would really advise making the concept clearer. If it is too vague, as I feel it is now, then it becomes very subjective – in fact, too subjective, because anything can then become a severe mental illness.
In general, I also feel the introduction could perhaps be clearer. For example, paragraph 1 is on the concept of severe mental illness, but starts of, to me atleast, somewhat confusing: “Severe mental illnesses (SMI) include different psychiatric disorders that impede global functioning, and require care and/or treatment on a long-term basis. Deficient functioning of persons with a SMI (pSMI) could (re)induce psychiatric symptoms, resulting in a circular, symptom-enhancing process”. First, I’m not sure how those two sentences really relate to each other. Second, would it not be better to first explain what an SMI is before you state that it needs treatment. Especially considering that you end that paragraph with again stating that treatment is necessary: “require care and/or treatment on a long-term basis” and then “In general, intensive care coordination or network- 45 based community care for pSMI is recommended”.
“Finally, both stigmatization and self-stigmatization increase barriers to proper physical healthcare[10, 11, 12].” It somewhat feels that this sentence is just put here because it found no place at another paragraph, despite that you already have sentences on care.
“Importantly, the use of psychotropic medication negatively influences many of the factors mentioned” (51-52). Isn’t it then better just to add after you mention those factors?
My point is: what is stated is not wrong, but it does seem as if the sequence of sentences does not feel entirely correct. The flow of the text is really strange at times, so I would suggest looking into that – it’s not the language itself, it is the structure.
Methods
“Based on grounded theory”. Yet, the result section does not look anything like a grounded theory – especially considering you did not make a theory.
Also, would that not really muddle your results, to adjust your questions according to the results you have. Unless you wanted to make a theory. Which you don’t. In fact, in the way you describe it now, it actually means that there is a sort of confirmation bias. You find a few themes through your first interviews, and then, as you describe it, you wanted to find confirmation for those themes, despite that it was based on only a few people. It also seems as if there was no clear objective for this study. To be clear: in a half-structured interview guide study you will also take into account what you know from previous interviews. That is simply how it works, because you cannot ignore the knowledge you have gained.
I generally don’t feel like “doing interviews” needs a specific analysis model, but it’s odd that the authors themselves start mentioning specific analyses, but actually don’t expand on it. What analysis did you use then? It seems very much so that you used thematic analysis. However, you also use some of grounded theory… but actually not. It seems quite chaotic how it is described now. Just give it a name, and describe it accordingly. Now, you somewhat state what you have done, and then use some technical terms.
Then, you give us three questions. But how were these then adjusted according to the “grounded theory”?
Discussion
What I don’t understand: isn’t the healthcare system in Belgium quite divided? The study took place in Flanders, but a general statement about Belgium is made, so maybe if there is a sort of divide in powers between regions and federal level, expand on that somewhere. Can the statements in this study be applied to the whole of Belgium?
“Yet, importantly, due to the diversity among the researchers (nurse, psychologist, psychiatrist, GP in training), we may have reduced the risk of interpretation bias”. A couple of remarks: this should be mentioned in the methods, if the authors believe it to be important. Furthermore, the argument could also be made that this actually led to bias because of the power differences between the researchers: a psychiatrist is higher in social hierarchy than a nurse and GP in training. You should explain why such diversity is important – I know it’s a standard sentence for qualitative research, but I feel like nobody really knows anymore why they actually state it.
“A multi-disciplinary guideline regarding this target group therefore can be valuable in supporting healthcare professionals. International consensus highlights the necessity in providing a clear framework in addressing physical health in pSMI, of which shared care planning is an essential building block [24, 25, 26].” There is something I do not understand about this. First, the author states that “despite … various guidelines” that physical healthcare delivery is suboptimal. So, so solution is… More guidelines? That seems like a recommendation that could be better. How are you going to ensure that it is going to be used? I do not know about your field, but I can say from my own experience that guidelines exist – they really do, and that basically most of what they do, meaning that there is little use of them in practice, because guidelines are difficult to implement in different contexts.
Furthermore, I would assume these are mainly made up of expert opinions – ignore this if that is different for these guidelines – so, why would it be so good to follow guidelines if they based mainly on the experiences of experts?
“Essential point to follow-up in pSMI are well-described in in existing research [30, 31].” I don’t understand this sentence – grammatically speaking. (also, two times “in”)
Line 367-373: your recommendation is thus not a recommendation? That is somewhat odd and is a straw man argument.
I also don’t understand why you “also” refrain – what else did you refrain?
I feel like the discussion could be a lot better and it feels somewhat messy at times. For example: “Yet, importantly, due to the diversity among the researchers (nurse, psychologist, psychiatrist, GP in training), we may have reduced the risk of interpretation bias. Despite these limitations,”
So, you end your limitations with a strength, and then state “despite these limitations”. It is a small thing, but you kind of feel it when reading the discussion that it seems less polished.
The discussion also kind of feels superficial, considering how much data you have. For example, returning to the issue of guidelines. You state that a multi-disciplinary guidelines would be valuable. You also state in the result section: “Strict implementation of guidelines is not always successful due to resistance or lack of motivation on behalf of the patient, implying a tailored”. So, how do you want to bridge that gap then?
(also, another example of a messy sentence: “implying a tailored” what?)
Conclusion
Do my eyes deceive me, or is your conclusion in a smaller fontsize?
“Future research should focus on the development and evaluation of guidelines in the Belgian context, both on the level of patient outcomes and economically” – I would not add “future research” to your conclusion. But also, I feel that this sentence says a lot clearer what you wish for future research than in your discussion. I had to go back and look, and yes, you kind of say that, but it’s somewhat unclear in the discussion, to be honest, that you find this so important for further research.
I also feel most of this conclusion is not a conclusion of your study. “Community based mental health services in Belgium only emerged the past decade” – okay, but that’s not your finding, though, right? If it’s so important, why is that not even mentioned in the introduction? Why is it only mentioned in the conclusion, even in the abstract conclusion?
I think the conclusion really does not feel like a conclusion. It is really odd to read around I guess almost 10 pages of results and then get a conclusion that is 50% not about your study.
General smaller issues
There are some small punctuation errors. E.g. “of which 8in the focus group, a” (line 129) or “factors such ashigh no-show rates” (line 352). I kind of assume this is due to the formatting to pdf, but if not, please correct, because the text is full of them (especially the discussion)
There are also some sentences that are grammatically quite odd. I have given a few examples throughout this review.
Also, throughout the results, the language is sometimes not very scientific. “The financing of somatic care for mental health patients in Belgian health settings is found to be profoundly inadequate”. Profoundly? That is quite biased. Just say “inadequate”, because adding an adjective to it, makes it seem like you are the one stating that, not your respondents.
A tip of Mark Twain, that ironically applies more to scientific writing than to literary writing, is “Substitute 'damn' every time you're inclined to write 'very;' your editor will delete it and the writing will be just as it should be.” Adjectives like “profound” fall under that too – especially in a result section.
“The paucity of professionals and materials addressing physical health care is the worst in mental health community or outreach teams.” First, “in the mental health community”, second, that’s something they stated? Because you can’t make that statement if they didn’t state it.
I am also unsure if some of the self-references are necessary, and then mainly “Martens N, Destoop M, Goossens B, Dom G. [Somatic and pharmacological nurse care in 2b-teams in Flanders: a cross- sectional explorative study]. Tijdschr Psychiatr. 2018;60(6):374-85.»
Considering it’s in Dutch, does it not make more sense to use the study you performed in English as reference? As a far as I can see, the two references roughly conclude the same thing, no?
Author Response
We thank you for the received suggestions. Please find attached our replies on the suggestions and remarks that were made.
Kind regards.

Reviewer 2 Report
Introduction: The section has all the minimum elements, yet it needs to be expanded. I suggest adding a hypothesis and research question to allow readers to understand the paper better.
Methods: One potential weakness could be the recruitment strategy; the sample could be biased since only some of the physicians may be excluded just because they did not answer the invitation.
Also, needs to be more comprehensive in how to explain the complete strategy. I suggest a diagram or summary of how the methods and materials are structured.
Finally, the section is disorganized, and the reader gets lost easily. The tables are hard to read and need to be improved.
Conclusions: I think authors miss the opportunity to have a policy-related takeaway from the results. I would suggest to add more in this
Author Response

(The authors gave the same response as above.)

Reviewer 3 Report
the authors tried to use a relatively simple method to draw attention to an important deficiency in primary health care, namely the detection and basic care of mental health impairments. The draft is relatively short and easy to understand, which I think is an advantage. I accept that it would have been possible to present a preliminary hypothesis or to analyze the conclusions drawn based on the interviews in a more detailed manner in the discussion, but health policy decision-making can also rely on the findings and the solution of the identified problems in this form.
The authors could perhaps argue even more strongly in favor of strengthening mental health care, delete or correct potentially misunderstanding sentences, but I support the publication of the study.
Deterioration of mental health is becoming more and more common, presumably as a result of cumulative crises and their impact on individual lifestyles. Overburdened health care systems, and especially primary care, often ignore incipient mental disorders, but they cannot deal with more advanced forms either. This is why this research is important, and the conclusion can be drawn from it that the front line of health care must also be prepared for the care and treatment of pSMI. The results of the study are partly based on focus group qualitative interviews with doctors and patients, the methodology is simple, although it only covered a few people. The results and suggestions are clear, the question is whether resources and specialists are available to improve the situation. The special value of the draft is that it was written concisely and comprehensibly. I recommend it for publication.
Author Response

(The authors gave the same response as above.)

Round 2
Reviewer 1 Report
I thank the authors for the revisions.
Reviewer 2 Report
Thanks for the revised version. In my opinion, the only recommendation would be a native speaker check.